# Porous SiC and SiC/C_f_ Ceramic Microspheres Derived from Polyhydromethylsiloxane by Carbothermal Reduction

**DOI:** 10.3390/ma15010081

**Published:** 2021-12-23

**Authors:** Urszula Mizerska, Witold Fortuniak, Julian Chojnowski, Slawomir Rubinsztajn, Joanna Zakrzewska, Irena Bak-Sypien, Anna Nyczyk-Malinowska

**Affiliations:** 1Center of Molecular and Macromolecular Studies, Polish Academy of Sciences, ul. Sienkiewicza 112, 90-363 Lodz, Poland; wfortuni@cbmm.lodz.pl (W.F.); jchojnow@cbmm.lodz.pl (J.C.); srubin@cbmm.lodz.pl (S.R.); jzakrzew@cbmm.lodz.pl (J.Z.); sypieni@cbmm.lodz.pl (I.B.-S.); 2Faculty of Materials Science and Ceramics, AGH-University of Science and Technology, Al. Mickiewicza 30, 30-059 Krakow, Poland; nyczyk@agh.edu.pl

**Keywords:** silicon carbide microspheres, porous ceramics, carbothermal reduction, divinylbenzene cross-linker, free carbon content

## Abstract

A simple and inexpensive method for the preparation of porous SiC microspheres is presented. Polysiloxane microspheres derived from polyhydromethylsiloxane (PHMS) cross-linked with divinylbenzene (DVB) were ceramized under conditions leading to the removal of oxygen from the material. The content of free carbon (C_f_) in highly crystalline silicon carbide (SiC) particles can be controlled by using various proportions of DVB in the synthesis of the pre-ceramic material. The chemical structure of the ceramic microspheres was studied by elemental analysis for carbon and oxygen, ^29^Si MAS NMR, ^13^C MAS NMR, SEM/EDS, XRD and Raman spectroscopies, and their morphology by SEM, nitrogen adsorption and mercury intrusion porosimetries. The gaseous products of the thermal reduction processes formed during ceramization created a porous structure of the microspheres. In the SiC/C_f_ microspheres, meso/micro pores were formed, while in carbon-free SiC, microspheres macroporosity dominated.

## 1. Introduction

Silicon carbide (SiC) has outstanding mechanical properties and chemical stability at high temperatures [1,2,3,4] as well as remarkable electronic and optical properties [5,6]. For this reason, SiC ceramics have found practical applications in very harsh conditions [1,2]. Recently, porous SiC or SiC composite microspheres have gained increasing interest as new promising materials in many fields. Spherical particles confined in dimensions within the micrometer range exhibit superior properties to their bulk counterparts. They provide good flowability, high packing density and easy access to particle surfaces [7,8,9,10]. In addition, hierarchical porosity gives the materials a high surface area and facile infiltration of liquids to their pores [7]. Porous SiC-based microspheres could be used in many fields, such as: catalysis [11,12], hot gas filtration [13], thermal insulation [10], sensors [14], absorbers [15,16], molten metal filtering [17], membranes [18,19], and tissue engineering [20]. The interest has recently been focused on SiC composites as potential semiconductor materials for absorption of electromagnetic waves [21], anodes of lithium-ion batteries [22] and supercapacitors [23].

Polymer-derived SiC ceramics and their composites are obtained from various precursors, such as polycarbosilanes [7,24,25], polysilazanes [8,9,10,26], polysiloxanes and silicone resins [27,28,29,30,31,32]. Polysiloxanes are commercially accessible and less expensive than other SiC precursors. The method used to produce silicon carbide ceramics and their composites from polysiloxane precursors is based on the carbothermal reduction of silicon oxycarbide (SiCO) formed in the first stage of the polysiloxane ceramization [27,32,33]. As SiO, CO and CO_2_ gas products are generated in this process a porous structure of the ceramic material is formed [25,34]. A porous structure can also be produced by adding a dispersed sacrificial phase [7,31,33]. Carbothermal reduction requires a high temperature at which preservation of the spherical shape of particles may be a problem. A few years ago, we developed the synthesis of polysiloxane microspheres from polyhydromethylsiloxane (PHMS), which proved to be good precursors for the fabrication of SiCO ceramic microspheres [35,36]. Now we found that it is possible to produce spherical SiC particles by heating the modified PHMS derived microspheres for a long time in argon at 1600 °C until oxygen is removed from this material by carbothermal reduction. Thus, porous ceramic SiC microspheres containing a controlled free carbon content can be obtained from precursor polysiloxane microspheres in reasonable yield in one step. A significant mesoporosity and/or macroporosity, as well as high SiC crystallinity, may be developed in these spherical particles.

## 2. Materials and Methods

### 2.1. Materials

Polyhydromethylsiloxane (PHMS) of molar mass about 3000 product of ABCR(Karlsruhe Germany), 1,3-divinyl-1,1,3,3-tetramethyldisiloxane (DVTMDS) (ABCR, Karlsruhe Germany, 97%), poly(vinyl alcohol) molar mass Mn = 7.2 × 10^4^ (Avantor Performance Materials S.A., Gliwice, Poland) were purchased from producers and used without additional purification. Divinylbenzene (DVB) was purchased from Merck Life Science (Darmstadt, Germany) with a declared purity 80.1% is a mixture of 56/24 meta and para isomers and 18.8% of ethylvinylbenzene, Karstedt complex offered by Momentive Performance Materials (Leverkusen, Germany) 20 w% Pt.

### 2.2. Analytical Methods

^29^Si MAS NMR spectra were recorded using a DSX 400 Bruker spectrometer (Bruker, Billerica, MA, USA). Spectra were acquired in HPDec mode with a 4 μs pulse at 90° and with a 100 s delay between pulses.

^13^C MAS NMR, spectra were recorded using a DSX 400 Bruker spectrometer(Bruker, Billerica, MA, USA). Spectra were acquired in HPDec mode with a 5 μs pulse at 90° and with a 100 s delay between pulses.

Elemental analyzes of the preceramic siloxane microspheres for carbon and hydrogen content were performed using CHNS analyzer EuroVector model 3018 Elementar Analysensysteme (GmbH, Langenselbold, Germany).

Elemental analyzes of ceramic microspheres for the determination of free carbon, C_f_, were carried out in Łukasiewicz Research Network—Institute for Ferrous Metallurgy in Gliwice, Poland. C_f_ was calculated from C_total_ which was determined by the burning in oxygen of the studied sample mixed with PbCrO_4_ and the formed CO_2_ was analyzed in the Kulomat 702/SO/CS apparatus (Ströhlein, Germany).

Determination of the oxygen content in the ceramic microspheres was carried out using a gas analyzer model TCHEN-600 made by LECO Co.(LECO, St. Joseph, MI, USA) in Łukasiewicz Research Network—Institute for Ferrous Metallurgy in Gliwice, Poland.

Scanning Electron Microscope images were taken with a Jeol JSH 5500 LV microscope (JEOL Ltd., Tokyo, Japan) in a high vacuum mode, at the accelerated voltage of 10 kV. Samples were coated with a fine gold layer (about 20 nm thick), using sputter coater JEOL JFC 1200 (JEOL Ltd., Tokyo, Japan).

SEM–EDS spectra were recorded with the JEOL JSM-6010LA (JEOL Ltd., Tokyo, Japan) scanning electron microscope operating at accelerating voltage of 15 kV, equipped with an energy-dispersive X-ray detector. The surfaces of the analyzed samples were sputtered with a fine gold layer (20 nm thick) before analysis.

XRD diffractograms were obtained using a Panalytical X’Pert MPD PRO instrument (Malvern Panalytical, Egham, England) equipped with a roentgen lamp with copper anode, X’Celerator detector and Johansson monochromator. Measurements were carried out in the Bragg–Brentano geometry over a 2θ range from 5° to 75° with a 0.008° step and a step time of 106 s.

Raman spectra were recorded using a Raman HORIBA LabRAM HR micro spectrometer (HORIBA Laboratory, Kyoto, Japan) using an excitation beam at λ_exc_ = 488.0 nm.

TGA for analysis of free carbon content was performed by heating sample in air from 25 °C to 80 °C with rate of 5 °C/min and hold at 800 °C for 3 h using analyzer TGA 5500 (TA Instruments, New Castle, PA, USA).

Mercury Injection Capillary Pressure (MICP) measurements were performed in specialized porosimetric laboratory at AGH University of Science and Technology in Cracow, Faculty of Geology, Geophysics and Ecology. An AutoPore IV 9500 mercury porosimeter from Micromeritics (Micromeritics Instrument Corporation, Norcross, GA, USA) was used. The test procedure was developed based on a standard test protocol ASTM-D4404-1 (the American Society for Testing and Materials, Pensylwania, USA, 2010), and the instruction manual for the AutoPore IV measuring apparatus (AutoPore IV 9520 Operator’s Manual V1.09 2008).

The Micrometrics ASAP 2020 Plus gas sorption analyzer(Micromeritics Instrument Corporation, Norcross, GA, USA) was used to study the surface and pore size distribution of the prepared materials. The samples were degassed at 573 K until the pressure stabilized at 5 µmHg/min. Then, measurements of N_2_ adsorption–desorption isotherms were performed at the temperature of 77.159 K. The area and pore size distribution were calculated with the MicroActive software (for Windows). BET multi-point analysis (Brunauer–Emmett–Teller) was used to calculate the total area. DFT (Density Functional Theory) and BJH (Barret–Joyner–Halenda) models were used to determine the pore size distribution.

### 2.3. Preparation of Precursor Polysiloxane Microspheres

The fabrication of precursor polysiloxane microspheres is described elsewhere [37]. Description of the typical experiment as well as characterization of preceramic microspheres are also presented in Appendix A. Conditions of emulsification were the same for all samples PA-1 to PA-6 and PB-1 (speed 7000 rpm, time 90 s) with the exception of sample A-7 (speed 15000 rpm time 300 s). The size distribution of the pre-ceramic siloxane microspheres is shown in Appendix A.

### 2.4. Ceramization of Microspheres

Pyrolysis of polysiloxane microspheres at 1600 °C was performed in a flowing argon atmosphere at a flow rate of 100 L/h. An aluminum oxide vessel (DEGUSSIT^®^AL23) was loaded with a known amount of preceramic microspheres and was placed in a high-temperature tube furnace (Nabertherm RHTH 120/300/18, Nabertherm GmbH Germany). Heating rate was 5 °C/min. The samples were maintained for 5 h at the target temperature. Then, heating was switched off, and the sample was allowed to cool to room temperature in the flowing argon atmosphere. After cooling, the samples were weighed and analyzed. 

Oxidation of microspheres (sample A-7) to remove free carbon were performed in a Nabertherm furnace at the flow of air with heating from 20 °C to 800 °C with heating rate of 5 °C/min and hold at isotherm at 800 °C for 3 h.

## 3. Results

### 3.1. Precursor Polysiloxane Microspheres

It is well known that polysiloxanes can be thermally transformed into SiC ceramics. The process is complex and requires heating above 1200 °C [38]. Ceramization of the siloxane polymer in an inert atmosphere above 900 °C first produces SiCO material which contains nanodomains of free carbon, C_f_. Further heating above 1200 °C results in the segregations of SiO_2_ and SiC phases, Equation (1), which is accompanied by carbothermal reduction of SiO_2_ by the free carbon, Equation (2). If the quantity of free carbon is high enough, both processes can convert SiCO to SiC and produce gaseous products that remove oxygen from the material [38,39]. The excess of C_f_ remains in the ceramics.
SiC_n_O_m(s)_ ⟶ SiO_2(s)_ + SiC_(s)_ + C_f(s)_(1)
SiO_2(s)_ + C_f(s)_ ⟶ SiC_(s)_ + SiO_(g)_ + CO_(g)_ + CO_2(g)_(2)

Our goal was to find a suitable precursor polysiloxane microspheres and the right conditions of the ceramization process to obtain oxygen-free SiC material with a controlled C_f_ content, with relatively high ceramization yield and maintaining the spherical shape of the ceramic particles. The other aim was to explore the possibility of pores formation in SiC ceramics by controlling the formation of gaseous products in the carbothermal reduction process. Our previous studies showed that microspheres produced by aqueous emulsion processing of PHMS, retain their shape well in ceramization at high temperatures [35]. However, due to the relatively small amount of carbon in the polysiloxane, it is difficult to remove oxygen from the produced ceramics while providing a good SiC yield. The method to achieve this goal by other researchers was the introduction of various forms of carbon, such as charcoal [40], anthracite or graphite [27], carbon black or phenol resin [31,33], to the precursor. In our research, additional carbon was introduced to this material through the use of a large amount of a cross-linking agent, which was divinyl benzene (DVB). As demonstrated by Kleebe et al., PHMS cross-linked with DVB and heated to 1000–1450 °C gives SiOC material containing a large amount of aromatic free carbon [41,42]. We previously found that SiOC microspheres obtained from modified PHMS/DVB systems were distinguished by a high content of C_f_ [43].

Series of precursor polysiloxane microspheres were prepared using various proportions of DVB cross-linker (s.i. Appendix A). They were obtained by aqueous emulsion processing according to the method described elsewhere [35] in which the polysiloxane with partly grafted cross-linker dissolved in a water-miscible solvent is mechanically emulsified with water containing a surfactant. The synthesis of this precursor series and the detailed analysis of the chemical structure of the formed microspheres was published separately [37]. The ^13^C and ^29^Si MAS NMR analysis made it possible to evaluate their chemical composition. The characteristics of the fabricated microspheres are shown in Appendix A. This analysis showed that the chemical structure of these microspheres is largely influenced by the content of the cross-linking agent. A characteristic feature of these microspheres is a large number of reactive side groups on the polysiloxane chains, which are native SiH groups and SiOH groups formed during the synthesis of the microspheres. In addition, there are two types of bonds connecting the polymer chains, SiOSi bonds due to condensation of reactive groups of the polymer and Si-CH_2_CH_2_- bridges due to hydrosilylation of DVB (s.i. Appendix A). As the proportion of cross-linking agents increase, the number of SiOH groups and SiOSi bonds decreases, while the number of bridges formed by hydrosilylation increases significantly, Table 1. When the weight/weight ratio (*w*/*w*) of DVB to PHMS approaches 0.48, the content of SiOH groups, and thus SiOSi cross-linking, drops significantly. Under these conditions, the polymer is cross-linked mainly by DVB bridges. The influence of the *w*/*w* ratio of DVB to SiH on the structure of cross-linked microspheres is demonstrated in Figure 1, which compares the ^29^Si MAS NMR spectra of the microspheres obtained at low and high DVB content. This phenomenon leads to an increase in the carbon content in the preceramic microspheres from about 30 w% to 53 w% carbon as the DVB / PHMS weight ratio increases from 0.17 to 0.96, Table 1, Figure 2. In contrast, DVTMDS cross-linked microspheres, which do not have aromatic groups, contain less carbon. They are rich in hydroxyl groups and SiOSi bridges Table 1, sample PB-1.

These changes in the chemical structure affect the elemental composition of the precursor polysiloxane microspheres (Table 1) and strongly affect their behavior during the ceramization process, which will be discussed in the next section.

### 3.2. Ceramic SiC and SiC/C_f_ Microspheres

The precursor polysiloxane microspheres were subjected to ceramization in the atmosphere of argon at 1600 °C for 5 h. Micrographs of representative ceramic microspheres are displayed in Figure 3. The particles preserved spherical shape although their surfaces indicated the formation of macropores. The presence of macropores was confirmed by SEM, see Figure 3b,d,h. The ceramic yield for DVB cross-linked microspheres ranged from about 30 to 46.3%, while for the DVTMDS microspheres the yield was much lower, due to the small content of carbon, Table 2. A sharp increase in the yield of ceramics was observed when the weight ratio of DVB to PHMS was above 0.48, Figure 4. This increase in ceramic yield was accompanied by a change in color of the microspheres from beige-green to black, Table 2. The beige-green color is consistent with the presence of SiC [6], while the black color may indicate the presence of free carbon. These observations are consistent with the changes in the composition of the preceramic microspheres discussed earlier. Oxygen-rich siloxane particles are converted to amorphous SiC_n_O_m_ ceramics with various content of carbon when heated above 900 °C. Subsequent heating of SiC_n_O_m_ material to 1600 °C causes separation of SiO_2_, SiC and C_f_ phases, Equation (1), followed by a carbothermal reduction of SiO_2_ by the free carbon, Equation (2). However, when the amount of free carbon is low, it is completely consumed before all SiO_2_ is converted to SiC. Under these conditions, SiO_2_ reacts with SiC to form volatile SiO and CO according to Equation (3) [3,44,45]. The removal of oxygen from microspheres in the form of SiO significantly reduces the final ceramic yield. A particularly low yield is observed for the carbon-poor microspheres obtained from PHMS/DVTMDS, Table 3, sample B-1. The large amounts of gases released during the pyrolysis of these microspheres generate their high porosity, as described in paragraph 3.4 and crevices on their surfaces seen in Figure 3h.
2 SiO_2(s)_ + SiC_(s)_ ⟶ 3 SiO_(g)_ + CO_(g)_(3)

Obviously, the carbon-rich siloxane particles, which are formed when the *w*/*w* ratio of DVB to PHMS was above 0.48, are converted to SiC mainly by carbothermal reduction, Equation (2), leading to a higher yield of ceramic. These microspheres also contain residual free carbon.

The content of C_f_ in the SiC microspheres was determined by combustion at 800 °C in air by a thermogravimetric analyzer, TGA. The recorded maximum weight loss was taken as the content of free carbon, Table 2. The total carbon (C_total_) content was determined by combustion of ceramic microspheres in oxygen in a tube furnace at 1200–1300 °C using the coulometric detection method, Table 2. The C_f_ content was calculated from the mass balance assuming that the sample contains only SiC, SiO_2_ and C_f_, Table 2. Additionally, the elemental composition of the ceramic microspheres was confirmed by Scanning Electron Microscopy/Energy Dispersive X-ray Spectroscopy, SEM/EDS, (s.i. Appendix A). The obtained results are consistent with the TGA analysis and show that only ceramic microspheres were obtained from the aromatic carbon-rich siloxane particles, which are formed when the *w*/*w* ratio of DVB to PHMS was above 0.48, contain a significant amount of the free carbon, Figure 4. The aromatic carbon is also visible in the ^13^C NMR HPDec/MAS spectra, Figure 5. The spectra show a broad signal of sp^2^ carbon-centered at about 110 ppm belonging mostly to C_f_ and a sharp signal at 20 ppm of sp^3^ carbon originating mainly from SiC. The sp^2^ signal is enhanced by the NOE effect [46] thus the results have only qualitative meaning.

It is worth mentioning that polysiloxane microspheres obtained in our standard method [36] using 1,3-divinyltetramethyldisiloxane as crosslinker (Sample B-1) was subjected to the ceramization under the same conditions as other samples. The analysis showed that the ceramic SiC microspheres obtained in this way did not contain any C_f_ and the yield of the ceramization was very low, Table 2. These microspheres retained spherical shape and exhibited a high macroporosity, as it is discussed below.

The ^29^Si MAS NMR spectra of the ceramic microspheres shown in Figure 6 confirm the presence of SiC. The signal corresponding to SiO_2_ at −108 ppm was not detected. Signals of other oxygen-containing silicon tetrahedra were not detected either. These suggest that residual oxygen in the ceramic microspheres should be below 2 w%, which was confirmed by elemental analysis, Table 2. The only silicon species visible in these spectra are those originating from silicon carbide. It exists mainly in the crystalline form of β-SiC, which is confirmed by the presence of a sharp signal at −19 ppm. Broader signals ranging from −23 to −27 ppm can be attributed to the amorphous form of silicon carbide and other crystalline SiC polytypes.

Further information about the structure of the silicon carbide phase is provided by X-ray diffraction studies. The diffractograms for all samples are similar to each other and indicate the high crystallinity of the silicon carbide phase which appears as a β-SiC crystal structure. Results for representative samples are shown in Figure 7. The main diffraction peak, which is observed at 2θ equal 35.60°, is attributed to the cubic β-SiC (111) phase. Other diffraction peaks that appear at 2θ: 41.4° (200), 60° (220), 71.8° (311) and 75.5° (222) also originate from the 3C β-SiC phase. There are no peaks characteristic for hexagonal α-SiC phases at 43.3° and 45.3° [38,47]. The absence of a broad halo at 2θ between 15–30°, characteristic of the amorphous SiOC and SiO_2_ phases, also confirms that the residual oxygen content must be low.

Raman spectroscopy study provides information on the structure of the C_f_ phase. Representative spectra for microspheres with high amounts of DVB are displayed in Figure 8 and some results are summarized in Table 3. The spectra of microspheres cross-linked with smaller amounts of DVB (A-2 and A-3) are similar to A-1, they have no free carbon in their structures. The Raman spectra show a series of bands, the most prominent of which are disorder-induced band D about 1360 cm^−1^, a graphite-like band G around 1590 cm^−1^ and a D-overtone peak close to 2950 cm^−1^ which is also induced by disorder [48,49]. Two other bands are superimposed on the G band. The resolution of these overlapping bands using Gaussian–Lorenzian approach (Appendix A) allows the evaluation of ID/IG of the integrated D and G bands intensity ratio (Table 3) used to quantify the graphite network defects, which is a measure of the disorder in the C_f_ phase. The ratio is larger than 1, indicating a high degree of disorder in the C_f_ phase which forms graphitic-like domains. The average in-plane linear size of these domains (La) was calculated using the Tunistra–Koenig (TK) approach [50]. The obtained La values were within the range in which the validity of the TK approach was previously verified [51].

### 3.3. Porosity by N_2_ Adsorption

The microspheres had a hierarchical micro/meso/macroporous structure which was characterized by N_2_ gas porosimetry and mercury intrusion porosimetry. Nitrogen adsorption informs about meso and micropores, while mercury intrusion gives knowledge about macropores. The results obtained in the nitrogen adsorption studies are summarized in Table 4.

The proportions of the cross-linking agent used in the synthesis of the precursor particles have a great influence on the type of porosity of the SiC microspheres. Ceramic particles made of precursors obtained with a very high DVB content (A-6 and A-7), thus with a large content of C_f_, are distinguished by a high specific surface area, SSA, which were 263 m^2^/g and 347 m^2^/g for samples A-6 and A-7, respectively. Average pore sizes were small, 2.62 nm in A-6 and 2.15 nm in A-7. The isotherms, shown in blue and green in Figure 9 for samples A-7 and A-6, respectively, have a concave shape facing the pressure coordinate, showing high nitrogen uptake at relatively low pressure. This indicates the high adsorption potential of these microspheres, which is characteristic of microspheres with narrow pores. This behavior is often observed in microporous materials [52,53]. A significant content of micropores is manifested in a large adsorbed volume at almost zero relative pressure. From the pore distribution shown in Figure 10a,b, it can be concluded that besides micropores these are mesopores with a width of 2 to 5 nm which mainly contribute to the porosity of sample A-6. The isotherm hysteresis of this sample shown in Figure 9 reflects this mesoporosity. Only a very small fraction of the pore volume is occupied by macropores.

Microspheres that were produced using a smaller amount of DVB, sample A-4, give different isotherms similar to those of type IV, according to the IUPAC classification [52] typical for mesoporous materials, Figure 9. In contrast to the microspheres with the highest C_f_ content, isotherms are convex with respect to the pressure coordinate in the range of 0.5 to 1 relative pressure. High nitrogen uptake occurs over a larger relative pressure range, see Figure 9. A characteristic feature of these isotherms is the hysteresis loop associated with the occurrence of pore condensation. The presence of the micropore fraction is manifested in the very steep shape of the initial part of the isotherm. The inflection points appear at the same beginning, about 0.015 P/P_0_, so they are only weakly marked. The microspheres without free carbon, samples A-11–A-3 and B-1, are represented by isotherms characteristic of a macroporous material with nitrogen uptake at a high relative pressure of 0.9–1. These isotherms are similar to each other so only this one of sample A-1 is displayed in Figure 9. The differences in the porosity of the microspheres are manifested in comparison to the distributions of their pore width. Plots of the cumulative pore volume as a function of pore width and as a function of pore surface area for microspheres containing low (A-1) and high (A-6) content of C_f_ are shown in Figure 10a,b. The corresponding plots for other samples are presented in supporting information (Appendix A). The micro and mesopores account for only about 25% of the pore volume of sample A-1 but constitute more than 95% of sample A-6 as deduced from Figure 10a.

Data in Table 4 reveal the variation in the BET area and in pore width. Samples A-1–A-3, which are free of C_f_, show similar small BET surface area and similarly large pore width. Instead, there is high variability and a clear trend in the values of these parameters within C_f_ containing microspheres (samples A-4–A-7). The pore area increases with growing C_f_ content, while the pore width becomes smaller. Additional experiments were performed to closer explain this behavior. The sample A-7, the one with the highest content of C_f_, was subjected to heating in the air atmosphere for 3 h at 800 °C. Elemental analysis confirmed full removal of C_f_. N_2_ adsorption porosimetry disclosed isotherm characteristics for mesoporous material. This sample contained mostly mesopores and very few micropores. Its SSA value decreased from 347 to 121 m^2^/g. This result clearly indicated that the generation of micropores is associated with the formation of the C_f_ phase. The air-heated sample was subsequently treated with concentrated HF solution. The specific surface area was slightly smaller, but the volume and width of the pores were significantly increased, see Table 4. During the heating of the microspheres in air, a quantity of SiO_2_ was formed which was confirmed by ^29^Si NMR MAS spectroscopy (Appendix A) showing a broad peak at −110 ppm. Most likely the oxidation took place on the wall of pores where the SiO_2_ layer was formed. This layer was removed by HF treatment.

### 3.4. Porosity by Mercury Intrusion

The results of the mercury intrusion studies are summarized in Table 5 and differential intrusion as a function of pressure and pore size for selected microsphere samples are displayed in Figure 11.

Uptake of mercury into interparticle voids occurs in the pressure range of 0–200 psi for most microspheres samples. As demonstrated in Figure 11, the intrusion into this volume is distinctly separated from the penetration into the pores of the microspheres, which allows for determining the characteristics of the pores. The results of mercury intrusion are not comparable with the results of N_2_ adsorption because micropores and a substantial volume of small mesopores are not penetrated by mercury. On the other hand, larger macropores escape the measurement of N_2_ adsorption. Instead, mercury intrusion gives full information about macroporosity. The open macropores occupy a large volume of the microspheres obtained with a small amount of DVB cross-linker, thus not containing C_f_. Sample A-1 shows an average pore width of 83 nm with an average SSA of 24.1 m^2^/g. The open porosity of the sample, calculated from the pore intrusion and skeletal density, is about 50%, however, the true value must be greater as a significant volume of open micro and mesopores are not intruded. The skeletal density of this sample is 2.03 g/cm^3^ compared to 3.21 g/cm^3^ for pure SiC, which reveals a large volume of non intruded pores, a substantial part of them may be open. Sample A-3 obtained using a moderate amount of DVB shows a large number of open macropores in the relatively narrow pore width range of 50–120 nm. In contrast, the microspheres A-5 obtained with a large amount of DVB used in the synthesis of the SiC precursor show very little mercury intrusion and almost no macroporosity. The intrusion to microspheres A-6 and A-7 was too small to be measured with reliable precision. Although the N_2_ sorption measurement of these microspheres gave a very high BET surface area but the average diameter of pores was very small. Their probable bottle-type shape may additionally hinder the mercury intrusion. 

## 4. Conclusions

We demonstrated that porous SiC and SiC/C_f_ ceramic microspheres can be obtained by a simple and inexpensive method from PHMS cross-linked with DVB polysiloxane microspheres. The polysiloxane microspheres were converted to the ceramic material by pyrolysis under conditions where thermal reduction removes oxygen from the ceramics. The weight ratio of DVB to PHMS used in the synthesis of preceramic particles controls the final composition of the ceramic microspheres. Ceramization of the siloxane microspheres obtained at a DVB/PHMS weight about a ratio higher than 0.5 leads to SiC material containing free sp^2^ carbon SiC/C_f_. SiC microspheres without C_f_ are obtained at the DVB/PHMS weight ratio of less than 0.5, but at the expense of ceramic yields. In this case, some oxygen remains after all C_f_ has been used up in the carbothermal reduction process. This oxygen is mainly removed as gaseous SiO formed by the reaction of SiO_2_ with SiC, which reduces the yield of ceramics. The thermal reduction process causes the formation of porosity in the ceramic microspheres, the nature of which depends on the content of the cross-linking agent in the preceramic particles. The SiC/C_f_ micro/mesoporous material is produced from siloxane particles obtained at a higher DVB/PHMS weight ratio. On the other hand, hierarchical macro/mesoporosity is generated in SiC microspheres obtained from siloxane microspheres obtained with a lower DVB content. The ceramic SiC phase mostly has a β-SiC crystalline structure, while the free carbon phase is in the form of a disordered graphitic structure.

## Figures and Tables

**Figure 1 materials-15-00081-f001:**
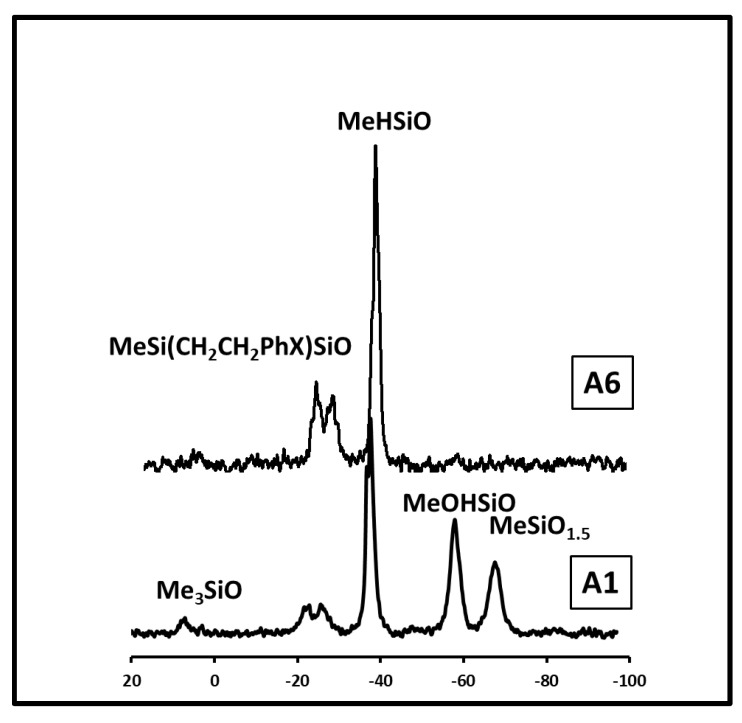
^29^Si MAS NMR spectra of the precursor polysiloxane microspheres obtained at different DVB/PHMS *w*/*w* ratio: A-1 = 0.17, A-6 = 0.97; X = CH_2_CH_2._

**Figure 2 materials-15-00081-f002:**
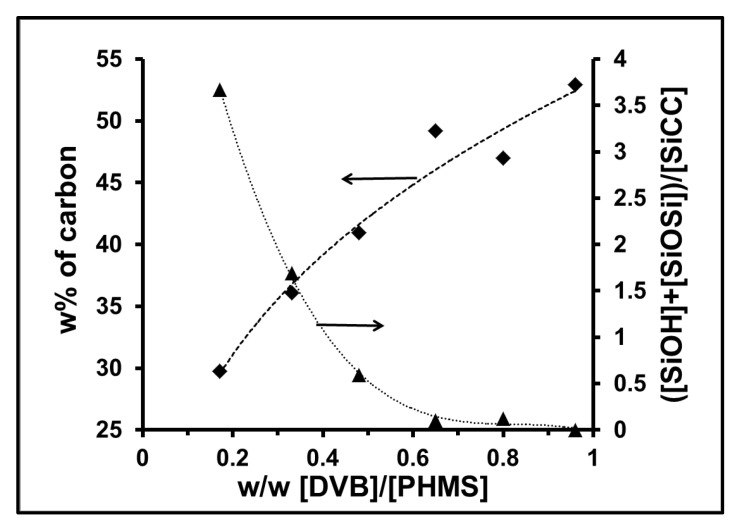
Effect of DVB proportion on the ratio of ([SiOH] + [SiOSi])/[SiCC]—▲ and the carbon content in the precursor siloxane microspheres—♦.

**Figure 3 materials-15-00081-f003:**
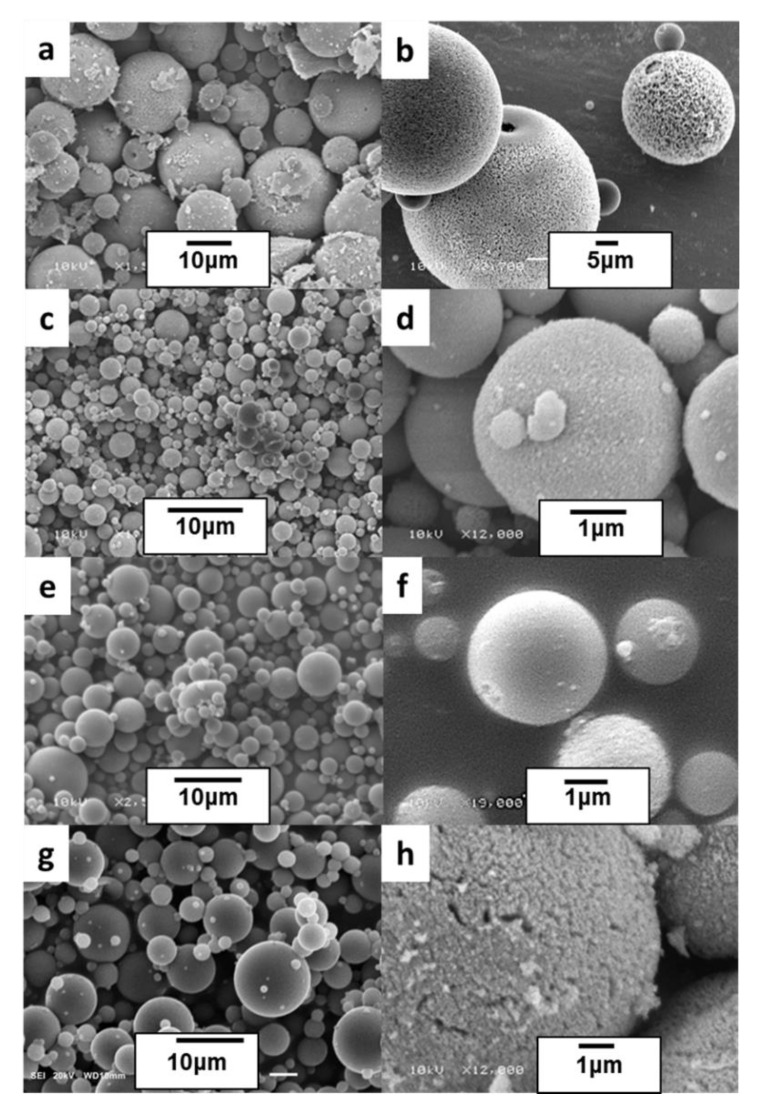
SEM images of SiC and SiC/C_f_ microspheres representative for different DVB/PHMS *w*/*w* ratio: (**a**,**b**) (A-1) = 0.17, (**c**,**d**) (A-3) = 0.48, (**e**,**f**) (A-6) = 0.97, (**g**,**h**) (B-1) DVTMDS/PHMS = 0.22.

**Figure 4 materials-15-00081-f004:**
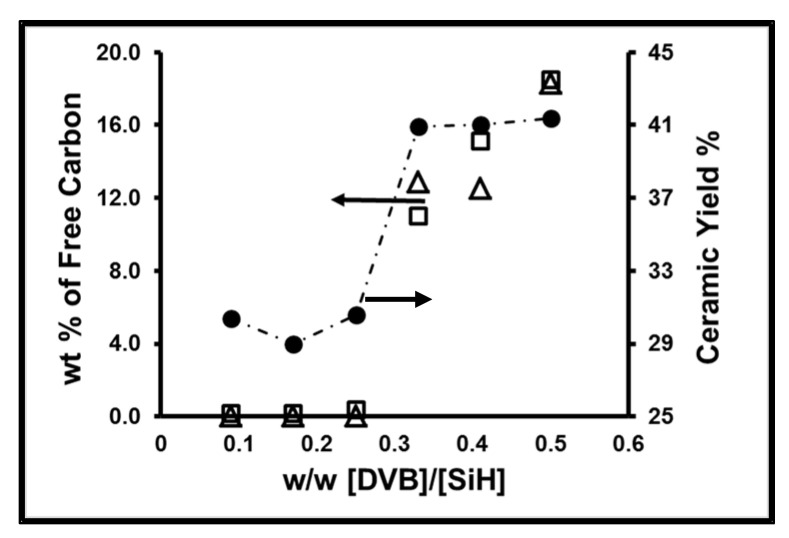
Effect of DVB proportion on free carbon content from (∆) elemental analysis and (□) TGA weight loss and (●) ceramic yield of ceramic microspheres.

**Figure 5 materials-15-00081-f005:**
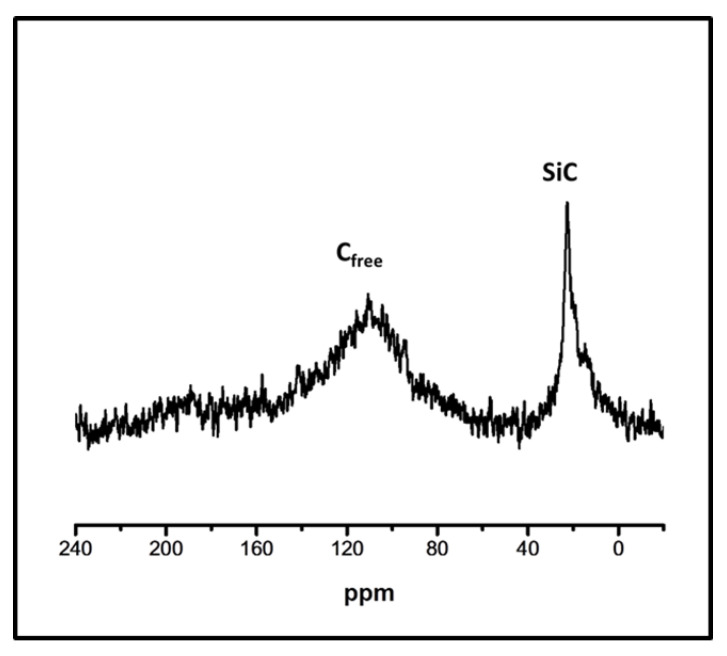
^13^C MAS NMR spectra of ceramic SiC/C_f_ microspheres obtained by ceramization of polysiloxane microspheres at 1600 °C for 5 h under argon; DVB/PHMS *w*/*w* ratio 0.97 (A-6).

**Figure 6 materials-15-00081-f006:**
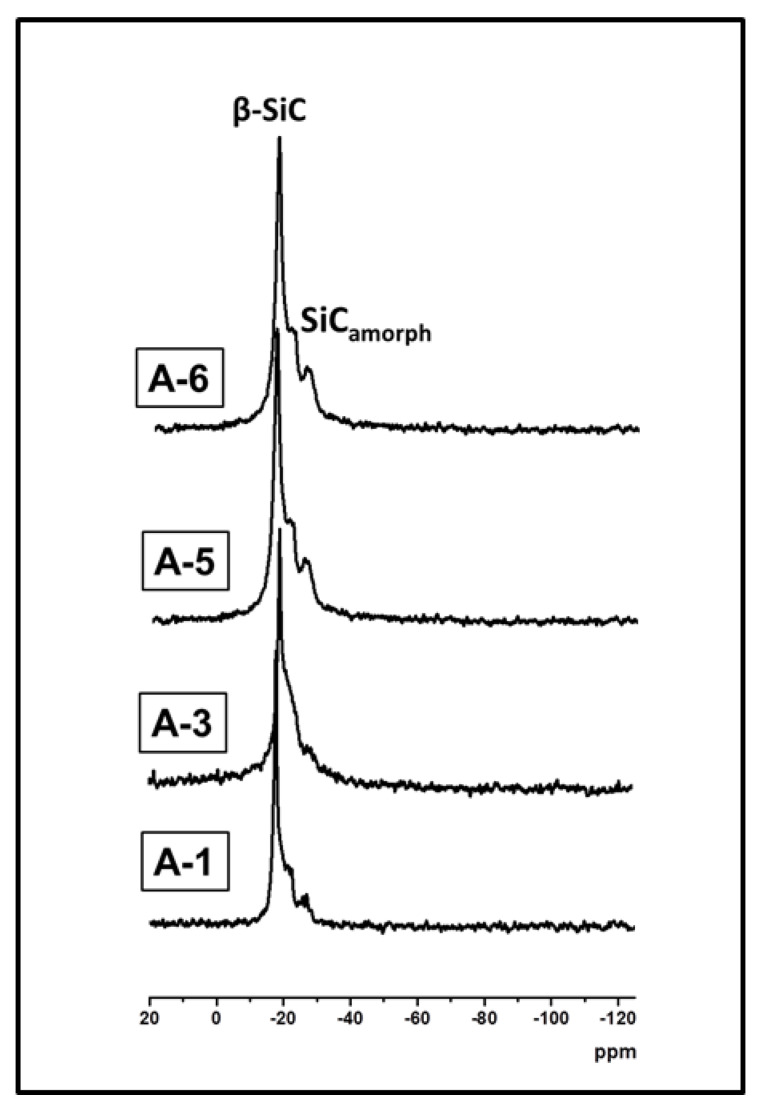
^29^Si MAS NMR spectra of SiC and SiC/C microspheres obtained at different DVB/PHMS *w*/*w* ratio: A-1 = 0.17, A-3 = 0.48, A-5 = 0.80, A-6 = 0.97.

**Figure 7 materials-15-00081-f007:**
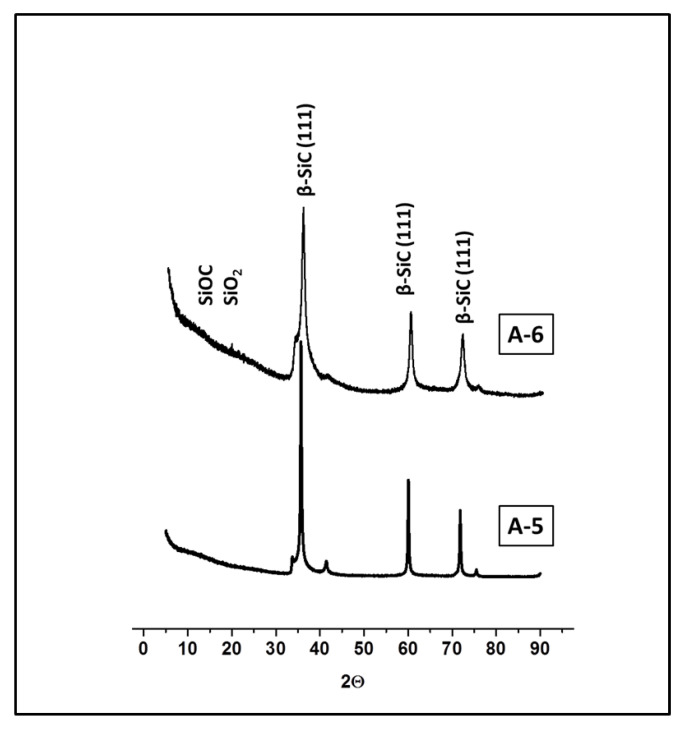
X-ray diffraction traces for SiC/C_f_ microspheres obtained by cross-linking with DVB.

**Figure 8 materials-15-00081-f008:**
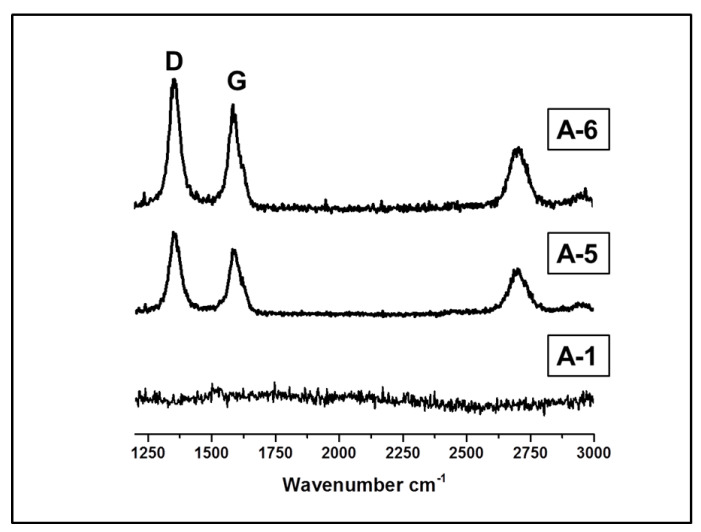
Raman spectra for SiC/C_f_ microspheres. DVB/PHMS *w*/*w* ratio: A-1 = 0.17, A-5 = 0.80, A-6 = 0.97.

**Figure 9 materials-15-00081-f009:**
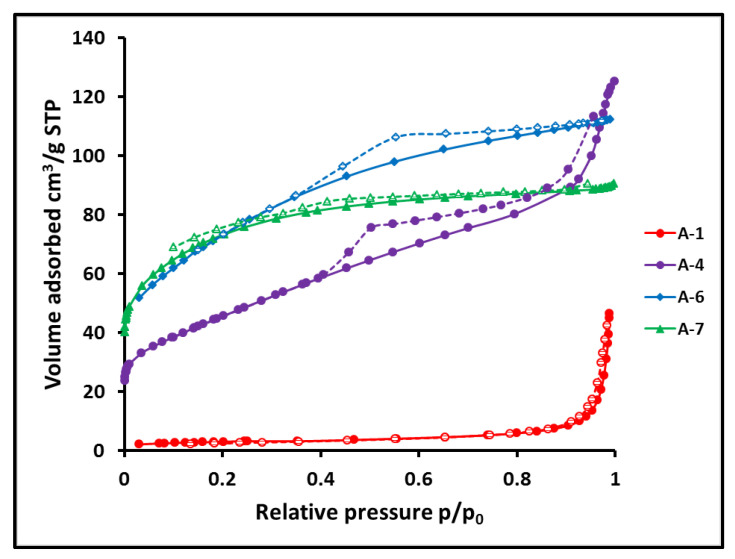
Isotherms of N_2_ adsorption of representative ceramic microspheres, DVB/PHMS *w*/*w* ratio: A-1 = 0.17, A-4 = 0.65, A-6, A-7 = 0.97.

**Figure 10 materials-15-00081-f010:**
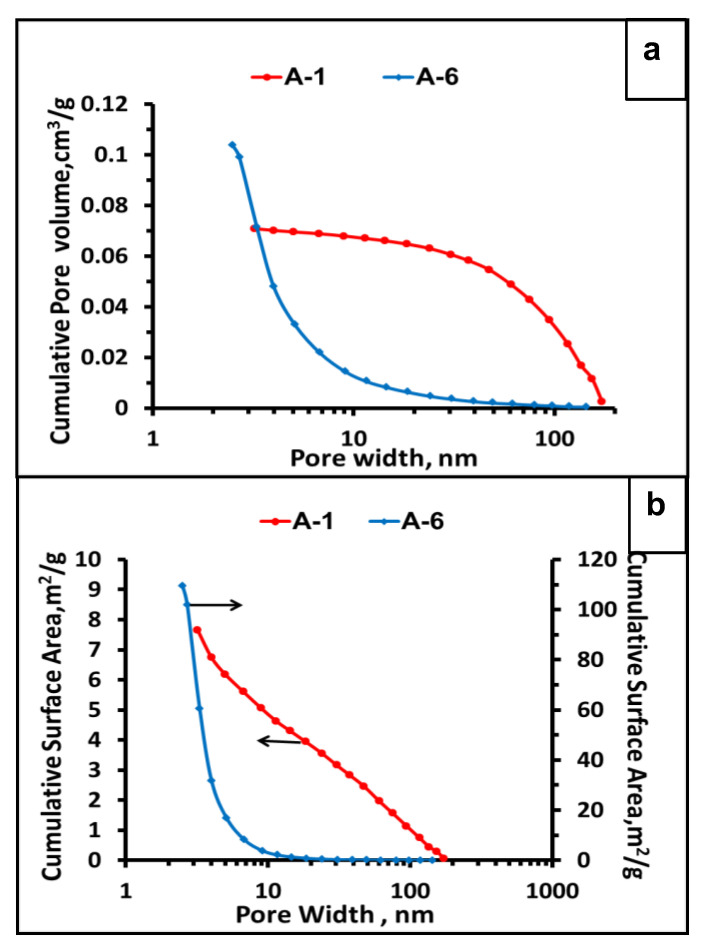
Cumulative plots of the distribution of pore width for samples A-1 and A-6: (**a**) by pore volume, (**b**) by pore surface.

**Figure 11 materials-15-00081-f011:**
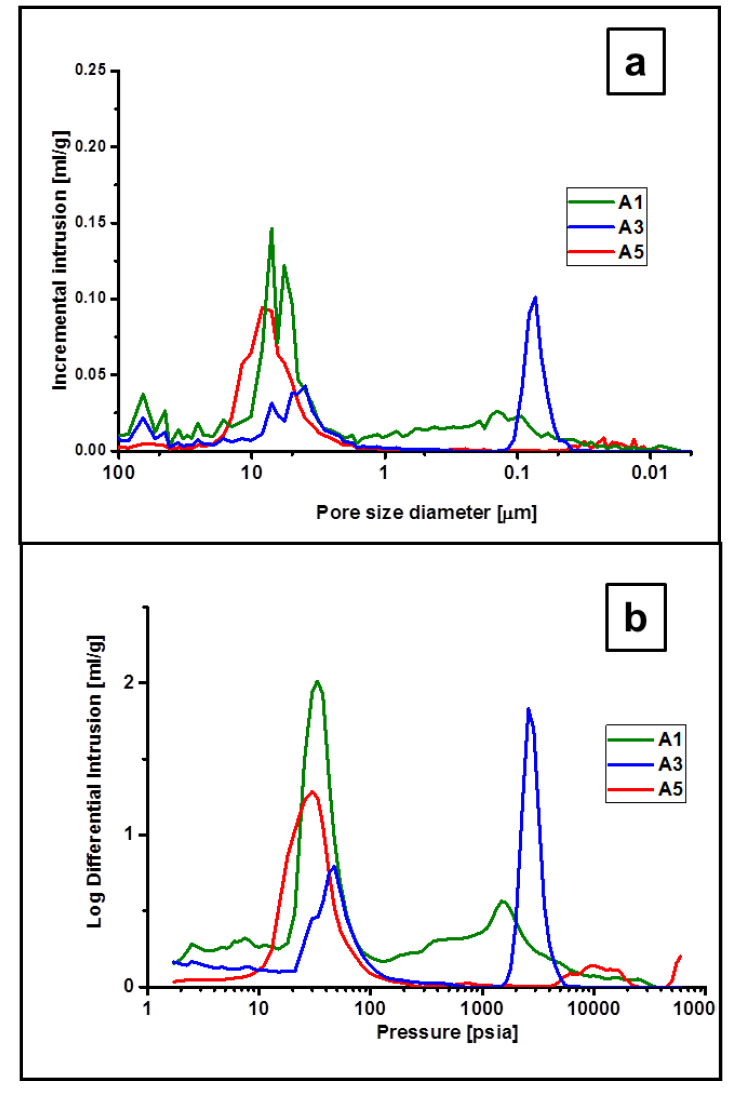
Mercury intrusion plots: (**a**)—Incremental intrusion vs. pore size, (**b**)—Log differential intrusion vs. pressure.

**Table 1 materials-15-00081-t001:** Carbon and hydrogen content and the ratio of ([SiOH] + [SiOSi])/[SiCC] in the precursor polysiloxane microspheres.

Sample	Crosslinker	Crosslinker/PHMS *w*/*w*	Av. C w%	Av.H w%	([SiOH] + [SiOSi])/[SiCC]
PA-1	DVB	0.17	29.8	6.03	3.67
PA-2	DVB	0.33	36.1	6.77	1.69
PA-3	DVB	0.48	42.49	7.11	0.59
PA-4	DVB	0.65	49.12	7.34	0.1
PA-5	DVB	0.80	46.98	9.83	0.12
PA-6	DVB	0.96	53.0	10.1	0
PA-7	DVB	0.96	55.6	9.44	0
PB-1	DVTMDS	0.22	22.67	6.42	5.29

PA 1-7 and PB-1denote preceramic samples corresponding to samples of ceramic materials A 1–7 and B-1.

**Table 2 materials-15-00081-t002:** Characteristics of ceramic SiC and SiC/C_f_ microspheres.

Ceramized Sample	Color	Ceramic Yield w%	Average Diameter μm	Weight Loss % TGA *	w% C_total_ **	w% O **	Calculated w% C_f_
A-1	Beige/Green	30.4	7.1	0.18	28	1.12	0
A-2	Beige/Green	29.0	3.5	0.17	28.2	1.54	0
A-3	Beige/Green	30.6	10.4	0.34	27.3	2	0
A-4	Black	40.9	11.1	11	38	1.85	11.4
A-5	Black	41.0	9.5	15.1	37.9	1.6	11.2
A-6	Black	41.4	2.1	18.5	42	1.45	17.1
A-7	Black	46.3	1.2	23.5	46.6	1.50	24.9
B-1	Beige/Green	8.5	4.5	0.38	30.2	1.06	0.2

* —ceramic microspheres were heated in air from 25 °C to 800 °C with rate of 5 °C/min and held at 800 °C for 3 h. ** —total carbon and oxygen content was determined in Institute for Ferrous Metallurgy, Gliwice, Poland.

**Table 3 materials-15-00081-t003:** Results of Raman studies of SiC/C_f_ microspheres.

Sample	D Position (FWHM) cm^−1^	G Position (FWHM) cm^−1^	I(D)/I(G)	L_a_(nm) (ʎ = 488)
A-4	1356 (55)	1590 (46)	1.55	2.8
A-5	1356 (46)	1588 (40)	1.18	3.4
A-6	1356 (52)	1589 (43)	1.47	3.0
A-7	1350 (53)	1593 (57)	2.04	2.2

**Table 4 materials-15-00081-t004:** Porosity results from nitrogen adsorption measurement.

Sample	Nitrogen Gas Adsorption			
	BET Surface Area m^2^/g	BJH Pore Volume cm^3^/g	BET Av. Pore Width nm	BJH Av.Pore Width nm
A-1	10.48	0.072	21.4	28.2
A-2	11.24	0.058	28.2	32.0
A-3	12.6	0.052	14.64	20.8
A-4	164.9	0.175	4.70	5.21
A-5	204.6	0.195	4.16	4.59
A-6	263. 5	0.145	2.63	3.1
A-7 *)	347.2	0.091	2.15	2.9
A-7-1 **)	120.7	0.218	8.41	10.94
A-7-2 ***)	103.0	0.347	15.81	15.63
B-1	53.01	0.281	20.30	21.47

Samples A-1–A-6 were obtained from DVB/PHMS using homogenization conditions: time 90 s, velocity 7000 rpm. *) Sample A-7 was obtained using the same DVB/PHMS ratio as in the synthesis of sample A-1–A-6, but at other homogenization conditions (time 180 s and velocity 15000 rpm). **) Sample A-7-1–A-7 was heated in air at 800 °C for 3 h. ***) Sample A-7-2–A-7-1 was subjected to the treatment with HF. Sample B-1 was obtained from DVTMDS/PHMS at the same conditions as samples A-1–A-6.

**Table 5 materials-15-00081-t005:** Characteristic of the microsphere’s porosity by mercury intrusion.

Sample	Total Intrusion (Pore Intrusion) cm^3^/g	Surface Area m^2^/g	Av Pore Width nm	Powder Density g/cm^3^	Skeletal Density g/cm^3^	Porosity %
A-1	1.53 (0.50)	24.1	83	0.495	2.03	50.5
A-3	0.829 (0.39)	19.5	78	0.715	1.76	32.0
A-5	0.764 (0.090)	25.6	13.6	0.725	1.63	12.8
B-1	1.33 (0.495)	43.2	46	0.560	2.20	52.4

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
