# Peer review of "Porous SiC and SiC/C_f_ Ceramic Microspheres Derived from Polyhydromethylsiloxane by Carbothermal Reduction"

_materials, 2021, doi:10.3390/ma15010081_

Round 1

Reviewer 1 Report

The manuscript is well organized and shows a well-executed research which deals with the synthesis of porous SiC particles based on the polymer derived ceramic PHMS and the divnylbenzene. The divylbenzene rate enables to control the carbon rate in the SiC and to control the porosity (macro, mesopore…).

In this article, many data are reported and comments on the characterization of particles obtained are often missing. The paper should be reworked in a more systematic way. I request major revision.

Few minor remarks:

Line 49: typographical error : a problem

Line 226: typographical error : “consumed”

Line 311: typographical error “cm-1”

Line 374 typographical error “microspheres”

Remarks and questions:

Materials and methods section

  • VTMDS is present in the whole document since line 68 but information is missing from the material section to know why it is mentioned. Indeed, the information comes a little late on lines 274-275. Could you provide more details in the materials section for DVTMDS and complete the information for ethylvinylbenzene?

  • Could you precise also synthesis conditions, like for PA-7 (line 339&340) time and velocity, for the synthesis for PA-1 to 6 and PB

  • Could you precise terms PA-X, A-X and B-X

  • Line 124: could you check the reference (35 or 37?) because p21, reference 35&37 are mentioned and place the reference between []

Results section

  • Following to the remark 1, could you add comments on DVTMDS results ex for Figure 3, Table 1 and for the other results
  • Following to the remark 2, PA-7 or A-7 comments should be added in the different sections for example in 3.2 and more precisely on the ceramic yield. Line 216 the ceramic yield is 42% but in Table2, ceramic yield could go up to 46.3%.
  • Line 222, as the work is done at 1600°C, this sentence is maybe not necessary.
  • Line 253 to 256_257: this part should be in analytical analysis.
  • What is the amount of silicon taken for the calculation in Table 2.
  • On Figure 8 could you add A-3 and comment or could you comment A-2, A-3 and confirm that no Cf are observed as previously shown
  • Figure 9, same question A-1 to A-3 have the same behavior? Could you add a comment?
  • Paragraph 3.4: could you comment A-2, A-4, A-6 and A-7 as this is done for BET, BHJ and add a comment on B-1

Author Response

The authors thank you for reading the manuscript carefully and for the numerous comments you have made to help improve this article.

VTMDS is present in the whole document since line 68 but information is missing from the material section to know why it is mentioned. Indeed, the information comes a little late on lines 274-275. Could you provide more details in the materials section for DVTMDS and complete the information for ethylvinylbenzene? --- DVTMDS (1,3-divinyl-1,1,3,3-tetramethyldisiloxane) is the cross-linking agent that was used to prepare the microspheres designated as B in the manuscript. The ceramic yield of SiC microspheres from DVTMDS is very low due to the removal silicone as SiO. They are only a reference for the aromatic carbon containing microspheres to show the differences. Some new explanations concerning DVTMDS was added to the sections 3.1 and 3.2 and under Table 1 and 4.   Ethylvinylbenzene is a DVB contaminant and was listed in line 72. We confirmed it in NMR, but we did not purify the DVB and did not extract it from the mixture. Ethylvinylbenzene does not interfere with the formation of microspheres, but introduces aromatic carbon.

Could you precise also synthesis conditions, like for PA-7 (line 339&340) time and velocity, for the synthesis for PA-1 to 6 and PB  - Conditions of emulsification were the same for all samples PA-1 to PA-6 and PB (speed 7000 rpm, time 90 s). This information is provided in Section 2.3. It was added to the text under Table 4.

Could you precise terms PA-X, A-X and B-X --- PA-X refers to DVB cross-linked pre-ceramic microspheres, A-X - DVB cross-linked microspheres after ceramization; B-X cross-linked DVTMDS microspheres after ceramization.  It was added to the text under Table 1.

Line 124: could you check the reference (35 or 37?) because p21, reference 35&37 are mentioned and place the reference between [] Results section ---Done in a manuscript

Following to the remark 1, could you add comments on DVTMDS results ex for Figure 3, Table 1 and for the other results --- The ceramic yield of SiC microspheres from DVTMDS is very low due to the removal silicone as SiO (Table 3). It was added to the text.

Following to the remark 2, PA-7 or A-7 comments should be added in the different sections for example in 3.2 and more precisely on the ceramic yield. Line 216 the ceramic yield is 42% but in Table2, ceramic yield could go up to 46.3%. --- Done in a manuscript

Line 253 to 256_257: this part should be in analytical analysis --- Done in a manuscript

What is the amount of silicon taken for the calculation in Table 2. --- The w% of silicon taken for the calculation in Table 2 was 100% – (w% Ctotal + w% O)   

On Figure 8 could you add A-3 and comment or could you comment A-2, A-3 and confirm that no Cf are observed as previously shown ---The Raman spectra for samples A-2 and A-3 confirmed that there was no free carbon in them. The comment was added to the manuscript (line 312).

Figure 9, same question A-1 to A-3 have the same behavior? Could you add a comment?---The isotherms of hysteresis of samples A-1 to A-3 look similar, they overlap and are therefore not shown in Figure 9. There is only a graph for A-1 as an example. Short comment is added to the text.

Paragraph 3.4: could you comment A-2, A-4, A-6 and A-7 as this is done for BET, BHJ And add a comment on B-1---The intrusion to pores of A-6 and A-7 were too small to be measured with reliable precision. The comment is placed in the text in Section 3.4. Although samples A-2 and A-4 have been investigated, however, the  amounts of the materials at our disposal were not enough to be sure of the precision of results.  

Reviewer 2 Report

Authors present a simple and inexpensive method for a preparation of porous SiC microspheres. This an interesting and meaningful research work. The manuscript is well prepared and organized.  The porosity and pore size have been characterized by Nitrogen gas adsorption and mercury intrusion. However, the detail of pores can be better understood  If the SEM images of the pores and the formation mechanism of the  pores with nanosize  can be present.  

The additional comments are as follow:
1. In present study, porous SiC microspheres was prepared by ceramizinng of Polysiloxane microspheres  derived from polyhydromethylsiloxane (PHMS). The dependence of the microspheres' composition, porosity on the recipe of precursors was investigated and analyzed.
2. The research is original and relevant in this field.
3. Present study provide a simple and inexpensive method to prepare  porous SiC microphere in comparison with published material.
4. Authors should give a detail analyzing on what is the formation mechanism of the pores with nanosize, and what cause the difference of the porosity among different samples.
 For further understanding the porous microspheres, the pores may need examined by  SEM or TEM with higher magnification.
5. The conclusion  mostly  consistent with the evidence and arguments presented.
6. The references are appropriate.

Author Response

The authors thank you for reading the manuscript carefully and for the numerous comments you have made to help improve this article.

Authors should give a detail analyzing on what is the formation mechanism of the pores with nanosize, and what cause the difference of the porosity among different samples.

 For further understanding the porous microspheres, the pores may need examined by  SEM or TEM with higher magnification. ---In this study we searched for the relationship between the composition / structure of microspheres and their porosity. We have shown that ceramized microspheres with a high DVB content have large amounts of free carbon and are distinguished by a large specific surface - they have mostly micropores. On the other hand, microspheres with low amounts of DVB are macro- and mesoporous. We also showed (samples A-7-1 and A-7-2) that the microporosity  is related to free carbon in microspheres. A detailed explanation of the mechanism of the formation of nanosized pores requires more research.

Reviewer 3 Report

The authors provided a simple and inexpensive method for a preparation of porous SiC microspheres. XRD, SEM, Raman spectra, TGA and et al were used to analyze the of characterization of the samples. The results were innovative and the manuscript was written well. But there were still some problems should be revised:

(1) In the section of “Introduction”, “the introduction should briefly place the study in a broad context and highlight why it is important” is unnecessary. Please delete it.

(2) Please check the grammar of the all manuscript. For example, in the line of 26, the authors described “It silicon carbide (SiC)…..”, “it” should be deleted.

(3) The introduction should be revised. In the line of 58, “In the present study we found that it is possible to….” should be revised. The materials results should not be listed in the “Introduction”. The highlights should be listed in the introduction.

(4) In the section of “2.3 Preparation of precursor polysiloxane microspheres”, the “(37)” should be revised as [37]. And the authors described “Description of the typical experiment as well as characterization of preceramic microspheres are also presented in supplementary materials, Table S1, Figure S1 and S2”, this sentence should be deleted in the manuscript.

(5) In the section of “3.1. Precursor polysiloxane microspheres”, please explain the reaction condition of Equation 1, 2 and 3 in detail by thermodynamics.

(6) In Table 2, the data should be revised unified. For example, the “ceramic yield” should be revised as “Ceramic yield” and the data of the ceramic yield of A-2 should be revised as 29.0.

(7) In Figure 3, the size of the SiC and SiC/Cf microspheres were different. Please explain the reason and mechanism for the difference of the size.

(8) For the XRD analyses, the reference code of SiC should be listed.

(9) In the line of 275, “see Ref.36” should be revised as “[36]”.

(10) In Table 4, the BET of the samples were different. The trend and the formation mechanism should be explained in detail.

(11) What is the main application the of the SiC microspheres? Which field is suitable for the using of this material?

(12) The authors described that EDS was used. But I could not find the EDS results in the Figures and Tables.

Author Response

The authors thank you for reading the manuscript carefully and for the numerous comments you have made to help improve this article.

In the section of “Introduction”, “the introduction should briefly place the study in a broad context and highlight why it is important” is unnecessary. Please delete it. ---Done in a manuscript  

Please check the grammar of the all manuscript. For example, in the line of 26, the authors described “It silicon carbide (SiC)…..”, “it” should be deleted.--- Done in a manuscript

The introduction should be revised. In the line of 58, “In the present study we found that it is possible to….” should be revised. The materials results should not be listed in the “Introduction”. The highlights should be listed in the introduction.---Done in a manuscript 

In the section of “2.3 Preparation of precursor polysiloxane microspheres”, the “(37)” should be revised as [37]. And the authors described “Description of the typical experiment as well as characterization of preceramic microspheres are also presented in supplementary materials, Table S1, Figure S1 and S2”, this sentence should be deleted in the manuscript.--- Done in a manuscript 

In the section of “3.1. Precursor polysiloxane microspheres”, please explain the reaction condition of Equation 1, 2 and 3 in detail by thermodynamics --- The thermodynamics of the high temperature conversion of SiCxOy to SiC has been extensively studied in the pass and is well documented in the literature. Including a thermodynamic discussion of the process (Equation 1, 2 and 3) in our manuscript would increase its size and be beyond its scope. The structure of the obtained microspheres results mostly from kinetics of processes 1-3 .

In Table 2, the data should be revised unified. For example, the “ceramic yield” should be revised as “Ceramic yield” and the data of the ceramic yield of A-2 should be revised as 29.0.---Done in a manuscript

In Figure 3, the size of the SiC and SiC/Cf microspheres were different. Please explain the reason and mechanism for the difference of the size. ---The size of ceramic microspheres shown in Figure 3 depends much on the size of preceramic microspheres which are formed in emulsion process. Many factors have impact on this size such as viscosity, surface energy, surface tension additionally modified by emulsifier, diffusion, chemical reactions and others . The size of microspheres is changing in ceramization when also many factors has influence on the particle size . It is just obvious that the distribution of the sphere sizes is observed and average size may be different in various samples. Our research effort was not focused on the explanation of these complex phenomena.

For the XRD analyses, the reference code of SiC should be listed. ---Done in a manuscript

In the line of 275, “see Ref.36” should be revised as “[36]”. ---Done in a manuscript

In Table 4, the BET of the samples were different. The trend and the formation mechanism should be explained in detail.---The discussion is placed in the text following Figure 10

What is the main application the of the SiC microspheres? Which field is suitable for the using of this material?---SiC microspheres are used as a specialty filler for high temperature ceramics, polishing abrasives and support for catalyst for high temperature reactions. Silicon carbide is used to coat friction surfaces operating at high temperatures, such as the sidewalls of engine cylinders, and as heat shields in spacecraft. 

The authors described that EDS was used. But I could not find the EDS results in the Figures and Tables---SEM-EDS studies were done, results are shown in Supporting information (Figure S3 and Table S2). They are consistent with results obtained by EM (Table 2).

Reviewer 4 Report

Dear Editor,

In this paper , based on  ceramization of PHMS cross-linked DVB, a economical method  for a preparation of porous SiC microspheres has been presented with a certain degree of innovation and practicalit. The content of free carbon in highly crystalline silicon carbide particles is controllable by vary proportions of DVB. The characterization is appropriate and the results is realiable. 

There are two questions, 

1.As shown in Fig.3 and Table 2, there is a obvious particle size difference between A-1 ,3,4and A 2,6,7.  Explain the reason.

2. According to Table 1 and 2,  when the ratio of DVB/PHMS is 0.17-0.48 the yield is stable, an explain for this is needed.  

There are inappropriate expressions as follows.

Line25 The introduction should briefly place the study in a broad context and highlight why it is important.
The labels in Fig.2 is missed.
Line 308 Raman spectroscopy studies provides should be ...study...
Line 310 The Raman spectra show should be shows.
Line 311 1590cm-1 should be corrected.

Table 4, the explanation ***)A-7-1 should be ***)A-7-2.

Author Response

The authors thank you for reading the manuscript carefully and for the numerous comments you have made to help improve this article

As shown in Fig.3 and Table 2, there is a obvious particle size difference between A-1 ,3,4and A 2,6,7.  Explain the reason. --- Similar differences are observed  between preceramic microspheres. They were prepared by emulsion process where many factors influence the sphere size. Perhaps a lower viscosity of PHMS/DVB with larger amounts of DVB is the most important factor which causes smaller size of microsphers A-6 and A-7. 

According to Table 1 and 2,  when the ratio of DVB/PHMS is 0.17-0.48 the yield is stable, an explain for this is needed ---Ceramization of the siloxane polymer in an inert atmosphere above 900 ° C first produces an SiCO material that contains free carbon nanodomains. Further heating above 1200 ° C results in segregation of the SiO2 and SiC phases accompanied by carbothermic reduction of SiO2 by free carbon. If the amount of free carbon is large enough, both processes can convert SiCO to SiC and produce gaseous products that remove oxygen from the material, with excess Cf remaining in the ceramic. Then the ceramization efficiency increases up to 47%. 

Line25 The introduction should briefly place the study in a broad context and highlight why it is important. ---This sentence was removed. 

The labels in Fig.2 is missed. ---Done in a manuscript

Line 308 Raman spectroscopy studies provides should be ...study... --- Done in a manuscript

Line 310 The Raman spectra show should be shows---It is correct 

Line 311 1590cm-1 should be corrected.---Done in a manuscript

Table 4, the explanation ***)A-7-1 should be ***)A-7-2. --- 

It is correct:

 **) Sample A-7-1 was created, when A-7 was heated in air at 800 °C for 3h

***) Sample A-7-2 was created, when A-7-1 was subjected to the treatment with HF

Round 2

Reviewer 1 Report

Thank you for taking into accounts the remarks.

No addition of modification are required